# Tumor Necrosis Factor-Like Weak Inducer of Apoptosis and Selected Cytokines—Potential Biomarkers in Children with Solitary Functioning Kidney

**DOI:** 10.3390/jcm10030497

**Published:** 2021-02-01

**Authors:** Hanna Nosek, Dorota Jankowska, Karolina Brzozowska, Katarzyna Kazberuk, Anna Wasilewska, Katarzyna Taranta-Janusz

**Affiliations:** 1Department of Pediatrics, Gastroenterology and Nutrition, University of Warmia and Mazury, 10-719 Olsztyn, Poland; hanna.nosek@wp.pl; 2Department of Statistics and Medical Informatics, Medical University of Bialystok, 15-295 Białystok, Poland; dorota.jankowska@umb.edu.pl; 3Department of Paediatrics and Nephrology, Medical University of Bialystok, Kilinskiego 1 st., 15-089 Białystok, Poland; karolina.brzozowska@udsk.pl (K.B.); katarzyna.kazberuk@udsk.pl (K.K.); annwasil@interia.pl (A.W.)

**Keywords:** chronic kidney disease, cytokines, solitary functioning kidney, tumor necrosis factor-like weak inducer of apoptosis

## Abstract

This study was performed to explore serum tumor necrosis factor-like weak inducer of apoptosis (TWEAK) and its dependent cytokines urinary excretion: monocyte chemoattractant protein-*1* (MCP-1) and regulated on activation, normal T cell expressed and secreted chemokine (RANTES) with their relation to the kidney function parameters in children with solitary functioning kidney (SFK). The study included 80 children and adolescents (median age 9.75 year) with congenital and acquired (after surgical removal) SFK. Serum TWEAK and urinary MCP-1 and RANTES levels were significantly higher in SFK patients (*p* < 0.05). The serum TWEAK was positively related to serum creatinine (*r* = 0.356; *p* < 0.001). Moreover, in SFK the receiver operating characteristic analyses revealed good diagnostic profile for serum TWEAK with AUC (Area Under The Curve)—0.853, uRANTES—0.757, and for RANTES/cr.: AUC—0.816. Analysis carried out to identify children with impaired renal function (albuminuria and/or decreased estimated glomerular filtration rate < 90 mL/min/1.73 m^2^ and/or hypertension) showed good profile for TWEAK (AUC—0.79) and quite good profile for uRANTES and RANTES/cr. (AUC 0.66 and 0.631, respectively). This is the first study investigating serum TWEAK and urinary excretion of MCP-1 and RANTES together in children with SFK. Obtained results indicate that TWEAK and RANTES may serve as potential markers of renal impairment.

## 1. Introduction

Congenital anomalies of kidney and urinary tract (CAKUT) are one of the most common defects seen in newborns, with occurrence from 1:500 to 1:2000 births [1,2]. Currently, due to the high availability of ultrasound, the initial suspicion of solitary functioning kidney (SFK), is diagnosed in about 60–80% of cases prenatally.

People with congenital absence of one kidney (unilateral renal agenesis) or loss due to disease or kidney donation have decreased renal mass which is associated with compensatory increase in glomerular filtration rate (GFR) of the other kidney. The clinical significance of the reduced number of nephrons was described over thirty years ago by Brenner et al. [3] in the theory of hyperfiltration. Persistent hyperfiltration and glomerular hypertension caused glomerular sclerosis, as a result of which the number of normal nephrons continued to decline. Kidney biopsy in adult patients with unilateral renal agenesis showed features of glomerular sclerosis and interstitial fibrosis. The reasons for the development of nephropathy occurring in some patients with a solitary functioning kidney have not been confirmed. The factors determining the degree of compensatory hypertrophy of a solitary functioning kidney and the severity of hyperfiltration have not been explained either.

The importance of the problem is evidenced by the fact that main causes of chronic kidney disease (CKD) in the pediatric population are congenital anomalies of the kidneys and urinary system, which account for approximately 50% of all cases. Due to the improvement of perinatal care and more effective treatment of children with CAKUT, more and more patients survive to adulthood and develop symptoms of CKD.

The tests assessing kidney function (GFR, endogenous creatinine clearance, serum creatinine concentration) available in routine diagnostics are still not perfect, and most importantly, they do not detect subclinical changes. Proper biochemical assessment of patients with kidney disease is extremely important from the clinician’s point of view. Progressive kidney damage is often asymptomatic or mildly symptomatic for a very long time. Therefore, only the early detection of this damage on the basis of laboratory tests can help to reduce the risk of chronic complications and slow down the progression of the disease. That is why the search for new biomarkers with high sensitivity and specificity for the assessment of early renal impairment is ongoing.

Recent years, there has been a growing interest in the role of a protein important in the diagnosis of inflammatory and systemic diseases with multi-organ involvement, such as the tumor necrosis factor-like weak inducer of apoptosis (TWEAK). TWEAK belongs to the tumor necrosis factor superfamily of cytokines [4]. It is a type II transmembrane glycoprotein composed of 249 amino acids (mTWEAK—TWEAK anchored in the membrane). In the kidneys, TWEAK is expressed both on the proper cells of the kidney (renal tubular cells, and mesangial cells) and cells of the immune system infiltrating the kidneys, for example some leukocytes (monocytes, T lymphocytes). Fn14 expression in healthy kidneys is low, but increases with its damage. Pro-inflammatory cytokines increase the expression of TWEAK receptor within two hours. The presence of Fn14 can be found on mesangial cells, renal tubular cells, and podocytes [5]. Stimulating the Fn14 receptors, the TWEAK cytokine induces an inflammatory reaction in the glomeruli and interstitium, leading to mesangial cell proliferation and chronic fibrosis, and consequently to the development and progression of chronic kidney disease.

In studies carried out on a mouse model, TWEAK, apart from causing a direct inflammatory reaction, also stimulated mesangial cells, endothelial cells and podocytes to secrete cytokines, including the monocyte chemoattractant protein-1 (MCP-1, CCL2) and regulated on activation, normal T cell expressed and secreted chemokine (RANTES, CCL5). TWEAK was also confirmed to be a promoter of non-inflammatory compensatory hypertrophy of the kidney after unilateral nephrectomy in mice [6].

Hence, the question whether, similarly to the mouse model, also in children with a solitary functioning kidney, the concentrations of the above-mentioned markers increase and whether there is a relationship between their levels and the progression of chronic kidney disease.

The aim of this study was to 1. Assess and compare serum TWEAK concentration and urinary excretion of MCP-1 and RANTES in patients with congenital and acquired solitary functioning kidney; 2. Test TWEAK and the cytokines MCP-1 and RANTES potential usefulness as biomarkers of renal impairment in children with solitary functioning kidney; 3. Determine probable cut-off points, which may be used in clinical practice in differentiation of SFK children with impaired renal function.

## 2. Material and Methods

The study included 120 children and adolescents with congenital and acquired SFK and healthy peers diagnosed and treated at the Department of Pediatrics and Nephrology, Medical University of Bialystok and the Department of Pediatrics, Gastroenterology, and Nutrition, University of Warmia and Mazury, Poland.

The study group (B group) was divided into 2 subgroups. The A group consisted of 54 children (33 males, 21 females) with congenital unilateral renal agenesis; N group—26 children (15 males, 11 females) with acquired (after surgical removal) SFK. Additionally, patients from the study group (group B) were divided into those with features of impaired renal function (albuminuria and/or decreased eGFR < 90 mL/min/1.73 m^2^ and/or hypertension)—this subgroup consisted of 52 children (65% of studied patients; 32 males, 20 females) and those whose kidney function was normal (normoalbuminuria, eGFR > 90 mL/min/1.73 m^2^, without hypertension)—28 children (35% of studied patients; 18 males, 10 females).

Inclusion criteria for the study group were: aged 1 month—18 years with confirmed single functioning kidney (kidney ultrasound examination, dynamic renoscintigraphy). The group N included children whose nephrectomy was a consequence of a congenital kidney defect or trauma.

Exclusion criteria were: presence of other organs chronic diseases, a recent (within 4 weeks) acute illness of any kind, as well as clinical or laboratory signs of infection (elevated C-reactive protein, procalcitonin, normal urinalysis), use of drugs that may affect kidney function, other abnormalities in kidney ultrasound.

The exclusion criterion in the N group was nephrectomy due to a kidney tumor, in order to exclude the influence of chemotherapy on the obtained results.

The reference group (K) consisted of 40 children (median age 6.54 year; 21 males, 19 females), appropriately matched according to sex and age. Subjects in the reference group were recruited among medical staff children, and among participants from the OLAF study [7].

Inclusion criteria for the reference group: children and adolescents aged 1 month to 18 years, born at term, with normal birth weight, in whom physical examination, blood and urine laboratory tests, and kidney ultrasound were normal. Children were in good overall health, without history of a recent (within 4 weeks) acute illness of any kind, and any history of chronic diseases. They were not taking any medications relevant to kidney disease. No data relating the risk of hypertension and other cardiovascular diseases, diabetes or gout were found.

Demographic and clinical data were assessed. In all children, careful clinical history, underlying comorbidities and physical examination were estimated. Blood pressure (BP) was measured using either a manual auscultatory or an automatic oscillometric device. High BP was defined as systolic BP (SBP) and/or diastolic BP (DBP) values above the 95th percentile adjusted for age, gender, and height. SBP and DBP load was calculated as the percentage of readings exceeding the 95th percentile for age, sex, and height percentile during each period. BP load analyses were conducted using 25% as the cut-off value [7].

Venous blood for biochemical tests (creatinine, urea, uric acid, TWEAK) was collected in the morning after an overnight fast. Laboratory tests in all patients were performed during routine diagnostics.

The estimated glomerular filtration rate according to Schwartz (eGFR) was calculated with use of the formula: eGFR = 0.413 × height in cm/serum creatinine in mg/dL [8].

Excretion of urinary albumin (albuminuria) was determined in the urine collected during a 24 h period. In younger children, due to the difficulty in collecting 24 h urine, the albumin/creatinine ratio in the morning urine sample (UACR) was assessed. Albuminuria was defined as a daily excretion in the range of 30–300 mg/24 h and UACR 30–300 mg/g creatinine.

TWEAK concentration was determined by commercially available sandwich ELISA immunoassay kit from MyBioSource Inc., San Diego, CA, USA.

Urinary levels of MCP-1 and RANTES (uMCP-1 and uRANTES) were determined using a commercially available immunoassay kits from Wuhan Fine Biotech Co., Ltd., Wuhan, Hubei, China.

The urine for the determination of the tested markers levels (uMCP-1, uRANTES) was collected from the first morning urine sample into a disposable container. The urine was centrifuged and specimens were stored at <−80 °C for up to 6 months. Urine samples were gradually thawed at room temperature prior to testing.

The study was approved by the Bioethics Committee of the Medical University of Bialystok (RI-002/137/2018) and the Bioethics Committee at the Faculty of Medical Sciences of the University of Warmia and Mazury in Olsztyn (No. 27/2017).

The statistical analysis was performed using the Statistica 12.0 PL computer program (StatSoft, Tulsa, OK, USA). The significance level of *p* < 0.05 was used in all tests.

The relation between estimated markers and baseline characteristics was assessed using the Spearman or Pearson correlation analyses.

A receiver operating characteristic (ROC) curve analysis was performed to determine the predictive value of estimated biomarkers as well as to define their optimal cut-off values.

Additionally, to search for the optimal cut-off point to differentiate patients with impaired renal function from those with normal renal function the method of Classification and Regression Trees (CART) was used.

## 3. Results

The demographic, anthropometric, biochemical parameters and estimated biomarkers in patients and control subjects are shown in Table 1.

The demographic and anthropometric parameters (age, weight, height) did not significantly differ among the groups (B vs. K). Higher values of systolic blood pressure were found in the group of children with SFK (group B) compared to the reference group (*p* < 0.05). Congenital unilateral renal agenesis was diagnosed at the median age 3.67 years (Q1: 0.58; Q3: 9.50); the median age at which the nephrectomy was performed was 0.91 years (Q1: 0.6; Q3: 2.0). Comparison of patients between groups with congenital and acquired SFK showed significantly higher values of systolic blood pressure in manual measurements in the group of children with unilateral renal agenesis (*p* < 0.05). No differences were found between A and N groups (*p* > 0.05).

Both, in the group with unilateral renal agenesis (A), as well as in patients after nephrectomy (N), there were no significant differences between female and male patients in the assessed demographic and anthropometric parameters (age, weight, height), laboratory parameters (creatinine, urea, albuminuria levels), and blood pressure (*p* > 0.05).

Serum creatinine levels differed between the congenital SFK (group A) and healthy subjects (K), but not between the nephrectomy (N) and the reference (K) groups.

Moreover in the group of patients with unilateral renal agenesis (A) and after nephrectomy (N), albuminuria was found, without significant differences between these groups (*p* > 0.05).

The serum concentrations of TWEAK in the study group (B) were much higher than in the reference group (*p* < 0.0001). Urinary excretion of MCP-1 and RANTES in B group, both presented in pg/mL and in values adjusted to creatinine (pg/mg cr.), showed significant increase in comparison to healthy peers (*p* < 0.05).

Comparisons of all assessed biomarkers between studied patients with congenital and acquired solitary functioning kidney and healthy peers are presented in Figure 1.

The median serum concentration of TWEAK in patients from group A was 486.09 pg/mL, and in group N it was 577.18 pg/mL. These values did not differ significantly between SFK groups (*p* > 0.05). The median serum TWEAK in the reference group was 164.94 pg/mL, and revealed significant difference between groups A vs. K, and N vs. K (*p* < 0.0001). Higher excretion of uMCP-1 (pg/mL) was demonstrated only in patients with unilateral renal agenesis (A) compared to the reference group (K) (*p* < 0.05). MCP-1 urinary excretion adjusted to creatinine (MCP-1/cr.) showed similar pattern between A vs. K groups (*p* < 0.05).

Increase in urinary RANTES (pg/mL) was found in patients with congenital SFK patients (A) compared to the reference group (K) (*p* < 0.01). Higher values of uRANTES were also found in the group of patients after nephrectomy (N) compared to the reference group (K) (*p* < 0.01). Similarly, the ratio of urinary RANTES to creatinine concentration (RANTES/cr.) was much higher in children with congenital (A) and acquired solitary functioning kidney (N) compared to the reference group (K) *(p* < 0.01).

Further tests were carried out to identify correlations of estimated markers with parameters of renal function. As shown in Figure 2, in univariate analysis, serum TWEAK was positively correlated with serum creatinine (*r* = 0.356; *p* < 0.001).

In additional evaluation we found that 52 patients (65%) from the study group (B) showed renal impairment defined with albuminuria and/or decreased eGFR <90 mL/min/1.73 m^2^ and/or hypertension. The presence of albuminuria was found in 47.5% of patients from the study group, (51.8% of patients with unilateral renal agenesis and 38.5% of patients after unilateral nephrectomy). In total, 55% of all patients in the study group had eGFR <90 mL/min/1.73 m^2^, including 50% in the A group, and 65.4% in the group with acquired SFK. Hypertension was found in 15% of children from the study group (14.8% of children with congenital and 15.4% of children with the acquired SFK). Interestingly, higher TWEAK serum concentrations were observed in studied children with albuminuria and normal eGFR in comparison to control individuals with comparable eGFR and non-albuminuria (median 512.23 pg/mL, Q1–Q3: 389.09–731.51 pg/mL, *p* < 0.001).

ROC analyses were performed in order to assess the diagnostic efficiency of evaluated biomarkers (TWEAK, MCP-1, RANTES) in identifying children with SFK among all examined children (Table 2A), which revealed good diagnostic profile for serum TWEAK with AUC—0.853, uRANTES—0,757, and for RANTES/cr.: AUC—0.816. Analysis carried out to identify children with impaired renal function (albuminuria and/or decreased eGFR <90 mL/min/1.73 m^2^ and/or hypertension) among studied patients (Table 2B), showed good profile for TWEAK (AUC—0.79) and quite good profile for uRANTES and RANTES/cr. (AUC 0.66 and 0.631, respectively).

Furthermore the CART classification tree method was used for the differentiation of patients with impaired renal function from those with normal renal function.

Figure 3: The *CART* showed that the most optimal cut-off point of the serum TWEAK was value of 395.628 pg/mL. Patients with measurements above this value can be classified as participants with impaired kidney function, as in the study group (B) 76.6% of patients with TWEAK levels above 395.628 pg/mL presented with features of impaired kidney function (albuminuria and/or decreased eGFR < 90 mL/min/1.73 m^2^ and/or hypertension). It should be noted that when analyzing with ROC curve, a cut-off point of 421.934 pg/mL was determined (Table 2B); which is similar to the cut-off point proposed by the Classification and Regression Trees analysis.

In the case of uMCP-1, the process of creating a classification tree and searching for homogeneous classes leads to a large fragmentation of the study group. Therefore, it was not possible to emerge a clear division rule that could be applied in practice.

The optimal cut-off point for uRANTES excretion was 6.826 pg/mL. However, the diagram presented in Figure 4 showed that patients with uRANTES levels above 10.332 pg/mL were at risk of developing impaired renal function, and presented 80.95% of children with impaired renal function.

Importantly, according the ROC curve analysis the best cut-off value for uRANTES excretion was 10.734 pg/mL (Table 2B); in the analysis using the CART method, the proposed cut-off point is almost the same—10.332 pg/mL.

## 4. Discussion

Due to the increasing number of patients with chronic kidney disease, apart from tests aimed at detecting pathogenetic factors influencing the development of CKD, it is important to develop specific and sensitive diagnostic methods that will allow the detection of kidney damage very early, before the onset of clinical symptoms.

We can divide patients with solitary functioning kidney into two groups. The primary SFK results from congenital disorders of nephrogenesis, i.e., unilateral renal agenesis. The second group is called acquired SFK, resulting from postnatal renal loss due to trauma or as a result of therapeutic nephrectomy, for example, in patients with obstructive or reflux nephropathy.

For many years it has been believed that the absence of one kidney is a benign condition with no sequelae of proteinuria, hypertension or chronic kidney disease. This was due to the observation of adult living kidney donors, who, as assessed by Goldfarb et al. [9], 20–25 years after the donation of the kidney for transplantation, showed no significant sequelae in the form of proteinuria, deterioration of single kidney function, and hypertension. This was also confirmed by Gai et al. [10] in a literature review on living kidney donors. However, the study of Seeman et al. [11] showed that hypertension, proteinuria, and renal impairment were more frequent in children with unilateral renal agenesis than in the population of healthy children. In the group of children with SFK, both with congenital defects and after nephrectomy, Dursun et al. [12] demonstrated significantly higher serum creatinine concentration and lower eGFR compared to the reference group.

It is known that creatinine concentration depends on many external factors, such as gender, body structure, muscle mass, and diet. Importantly, increase in creatinine appears relatively late, when renal impairment is already severe. Furthermore, the presence of albuminuria and high blood pressure usually indicates advanced kidney damage. It was found that the loss of nephron mass (in the case of congenital or acquired SFK) leads to hyperfiltration of a single nephron and nephron hyperplasia. Hence, for a long time, we may not see an increase in creatinine concentration or observe albuminuria in the presence of SFK. Overload of individual nephrons leads to focal glomerulosclerosis and interstitial fibrosis what was confirmed in kidney biopsy of patients with unilateral renal agenesis [13,14].

Over the years, the search for markers of impaired renal function has been carried out, which would have a much greater sensitivity and specificity than creatinine, and would allow for subclinical kidney damage detection, differentiation between patients with mild to severe impaired renal function, and earlier implementation of therapies slowing the progression of the disease.

The discovered representative of the tumor necrosis factor superfamily—TWEAK seems to be a good candidate for the role of such a marker. It is still the subject of many studies, and its role, and importance in the course of various diseases in humans is still not fully understood.

It has been shown that TWEAK, by stimulating Fn14 receptors, directly induces an inflammatory response within the glomeruli and interstitium with the expression of various pro-inflammatory molecules, including MCP-1 and RANTES cytokines. It may stimulate angiogenesis and initiate the process of fibrosis [15,16]. In vitro, TWEAK has also been shown to have a proliferative effect on renal tubular epithelial cells growth and to influence on compensatory renal hypertrophy in mice undergoing unilateral nephrectomy [6]. TWEAK involvement in non-inflammatory kidney hyperplasia in an animal model raised the question whether similar changes occur in humans, and more specifically in children with solitary functioning kidney.

In the present study, we undertook the assessment of TWEAK and selected markers of fibrosis (MCP-1 and RANTES) in children with SFK, and their correlation with the parameters of kidney function. We tried to answer the question whether they might serve as potential indicators of early renal impairment.

Initially, the analysis included a comparison of biochemical parameters in studied children. Study by Schreuder et al. [17] conducted in 66 children with SFK showed similarly to our results significantly higher serum creatinine in comparison to participants with two normal kidneys. The presence of albuminuria (>20 µg/min) was found in 23% of patients, what is in disagreement with our results, but 17% of these patients had hypertension, which is consistent with the results of the current study. The prevalence of hypertension in our study was 15%. Hypertension was noted in 13% of cases by Westland et al. [1], 17% by Schreuder [17], and 26% by Dursun et al. [12]. In Radhakrishna et al. [18] study 91% of their study group had at least one of the markers of renal injury such as albuminuria, reduced eGFR, or hypertension. We also found that the major part of the study group had features of kidney damage (65% of the study population), which is more than found in other studies such as by Sanna-Cherchi et al. [19] (29.5%) and Akl (20%) [20].

In the largest retrospective KIMONO study [1,21], which included 407 children with congenital or acquired solitary functioning kidneys, 37% of respondents showed features of kidney damage (31% with congenital SFK and 45% with acquired kidney). These differences are probably due to different criteria for including patients in the group with decreased eGFR (our study eGFR < 90 mL/min/1.73 m^2^; in the KIMONO study eGFR < 60 mL/min/1.73 m^2^).

Then we analyzed the concentrations of the tested markers. In the available literature, we have not found data on the assessment of TWEAK concentration in children with SFK. Our study revealed higher serum TWEAK in the study group than in the reference group, and these differences were significant. Further, a separate analysis of TWEAK concentrations in group A and group N showed significant increase in TWEAK in group A compared to group K and in group N vs. K. However, there were no significant differences in the serum TWEAK between patient groups A and N. These results may indicate that an assessment of serum TWEAK levels could be useful in distinguishing patients with reduced nephron mass from those who have both normal kidneys. However, based on the evaluation of TWEAK values, it is not possible to differentiate the cause of the reduced presence of a solitary functioning kidney—congenital or acquired. Obtained results also raised the question if this molecule is accumulating due to eGFR decline? All of evidence suggests that TWEAK is involved in the pathological processes that occur locally in the kidneys. Indeed, it is unknown whether a relevant increase in the production of TWEAK or a severe reduction in renal filtration of TWEAK may increase their serum concentrations. Currently it is thought that TWEAK levels increase in patients with impaired renal function. However, up to now, very few data are available on the precise relationship between TWEAK and the level of GFR. In fact, in patients with severe decline in eGFR serum concentrations of different proteins are definitely increased, and as a consequence, the filtered load of these proteins to the residual nephrons may become higher than single-nephron maximal tubular reabsorptive capacity, increasing its urinary excretion. Interestingly, the urinary excretion of molecules begins from different threshold values of GFR and at different levels of serum concentrations, indicating that proximal tubular cells probably have a different reabsorptive capacity for the different proteins.

The largest amount of data from the literature regarding TWEAK levels concerns adult patients with lupus nephropathy. In 2007, Schwartz et al. in a cross-sectional, multicenter study [22] found that patients with lupus nephritis have significantly higher urinary TWEAK excretion than patients without renal involvement. The concentration of TWEAK in urine was already rising 4–6 months before the disease presentation. Many authors [23,24,25,26] have shown that urinary TWEAK excretion mirrors disease activity and correlates with other potential biomarkers such as MCP-1. This finding confirmed previous in vitro observations that TWEAK induces inflammatory mediators, known to be involved in the pathogenesis of lupus nephritis (MCP-1, RANTES) by stimulating murine mesangial cells [27].

Further results were obtained in terms of MCP-1 and RANTES urinary excretion (both in pg/mL as well as in pg/mg of creatinine). The couple of tested cytokines showed significantly higher values in the study group than in the reference group. A detailed analysis comparing the urinary excretion of the tested biomarkers in studied subgroups did not show significant differences in the urinary MCP-1 and RANTES between the unilateral renal agenesis group and the nephrectomy group. MCP-1 levels, both in pg/mL and expressed as pg/mg cr., were significantly higher in group A compared to the reference group. There was no difference in MCP-1 excretion between N and K groups. The analysis of RANTES revealed significant differences both between patients with unilateral renal agenesis and the reference group; as well as between nephrectomized patients and the healthy peers. Thus, it seems that the determination of RANTES concentrations in urine could be used to differentiate patients with SFK from those with two normal kidneys.

There is a little data in available literature on the assessment of MCP-1 concentration in the urine of children with a single kidney.

In study conducted in adults living kidney donors the authors concluded that MCP-1 concentration in urine may detect early tubulointerstitial fibrosis in adults with normal renal function determined with normal creatinine levels and absence of albuminuria, and can therefore be considered a non-invasive marker of renal fibrosis [28]. In another prospective study published by Bartoli et al. [29] in a group of 80 children with CAKUT (hypoplastic, agenetic, and nephrectomized due to CAKUT) increased levels of MCP-1 were demonstrated only in SFK groups. Above-mentioned results, in opinion of authors, are due to chronic renal inflammation.

Promising results of studies of this marker in adults with various kidney diseases, but also in children with urinary tract defects, suggest that MCP-1 could be a potential biomarker for the assessment of early kidney damage also in children with a single functioning kidney. Our study confirmed higher values of MCP-1 in SFK group, but there was no possibility to differentiate with its use between agenetic and nephrectomized patient.

The RANTES cytokine was also assessed as a potential biomarker of kidney damage in patients with systemic lupus erythematosus. In the group of 88 adult patients, Chan et al. [30] found a significantly higher urinary levels of RANTES. In another Chinese study, patients with lupus nephritis showed higher urinary RANTES and MCP-1 levels, however, only increased urinary RANTES levels seemed to be independent predictor of lupus nephritis [31]. We did not find reports regarding the use of urinary RANTES levels as a biomarker of kidney damage in children with a solitary functioning kidney.

Finally we tried to determine the cut-off points of the studied markers by analyses with the use of ROC curve and CART classification trees, which would allow us classifying children with SFK group with normal and impaired kidney function on the basis of the studied markers levels.

The ROC analysis determined the cut-off point for TWEAK equal to 421.934 pg/mL. Patients with a TWEAK levels above this value showed evidence of renal impairment, and the test specificity was 76.7%, and sensitivity 79.1%.

The analysis of CART classification trees showed that the most optimal cut-off point of patient differentiation is the TWEAK value of 395.628 pg/mL. Patients with serum TWEAK concentration above this value can be classified as impaired kidney function, because in the group with TWEAK > 395.628 pg/mL 77.60% of patients showed signs of renal impairment (albuminuria and/or decreased eGFR < 90 mL/min/1.73 m^2^ and/or hypertension). It should be noted that in both analyses the proposed cut-off point was similar.

For uMCP-1 concentrations, on the basis of the performed analyses, it was not possible to obtain the optimal concentration that would clearly indicate the presence of impaired renal function in patients with SFK.

In the case of uRANTES, the ROC analysis determined the cut-off point equal to 10.734 pg/mL. Patients with uRANTES above this value showed renal impairment, and the test specificity was 86.5%.

Based on the CART classification trees analysis, it was possible to determine the optimal division point—6.826 pg/mL. Among the patients with uRANTES levels above this value, 64.29% had impaired renal function. Further analysis with CART trees allowed the determine even more optimal level of uRANTES. The value of 10.332 pg/mL definitely differentiates patients into those with normal and impaired renal function, as 80.95% of patients with uRANTES concentration above 10.332 pg/mL showed features of impaired renal function. It is also very important that both the cut-off points in the ROC curve and CART trees are almost identical.

The results presented in this study confirm that in patients with unilateral renal agenesis and acquired SFK, increased serum levels of TWEAK as well as increased urinary RANTES excretion are observed in comparison to the reference group. Increased excretion of uMCP-1 occurs only in patients with congenital SFK. Unfortunately, no significant correlations between serum TWEAK concentration and urinary excretion of tested chemokines could be established. However, it was possible to confirm a significant positive correlation between serum TWEAK and creatinine concentrations.

All assessed markers achieved significantly higher concentrations in patients with SFK and features of impaired renal function, compared to those who did not show impaired function. Therefore, it is possible to use their determination to differentiate patients with a solitary functioning kidney with features of impaired renal function from those with normal kidney function. It should be mentioned that our study failed in differentiation between unilateral renal agenesis and acquired SFK with use of estimated markers.

We are fully aware of the limitations of the study, being small, single center and cross-sectional. Moreover, most of our patients were young children, an appropriate measurement of blood pressure in children is difficult because of “white coat” anxiety, widely varying arm size, and occasional poor cooperation. Furthermore, in prepubertal and younger children no formula of GFR estimation gives acceptable results. It is still controversial, whether in this population a correct estimation of GFR could be obtained from serum creatinine concentration. We still do not know which formula is the least misleading in younger children.

Further work is necessary to better define obtained results. For example, better standardization of methods for its measurement in serum and urine, evaluation of MCP-1, RANTES activity in serum, assessment of its protective role in renal injuries once damage has already occurred. Additional efforts comparing serum TWEAK concentration with its urinary excretion would be helpful for detailed answer the question of molecules accumulation due to eGFR decline.

In conclusion, what is extremely important, in the case of serum TWEAK and urinary RANTES it was possible to determine the optimal cut-off values, exceeding of which may indicate impaired renal function in patients with a solitary functioning kidney. It might be useful in clinical practice in establishing the reference values of new biomarkers in patients with solitary functioning kidney.

Reached results indicate that the tumor necrosis factor inducer, TWEAK, and the cytokine RANTES may serve as potential biomarkers of early renal impairment and suggest possible relationship between TWEAK and uRANTES levels and the degree of renal impairment. However, more studies on a larger group of patients are needed to confirm these preliminary data.

## Figures and Tables

**Figure 1 jcm-10-00497-f001:**
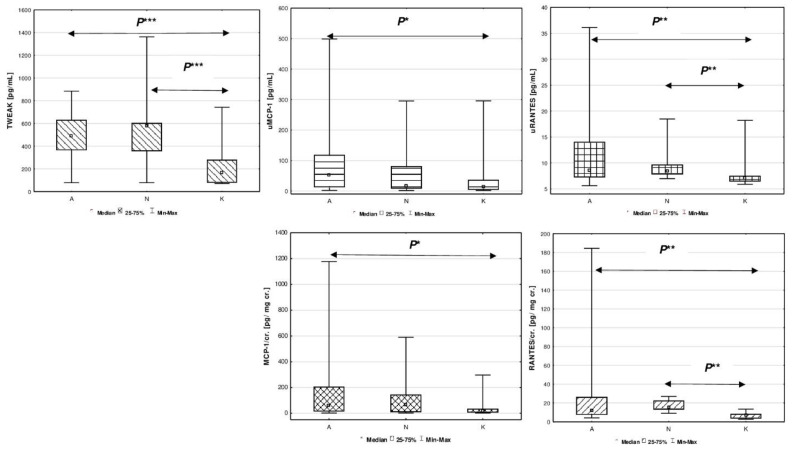
Comparison of estimated biomarkers between studied children with congenital A, acquired solitary functioning kidney N and the reference K groups. Figure legend: *P** *p* < 0.05, *P*** *p* < 0.01, *P**** *p* < 0.0001. (**A**) TWEAK [pg/mL]; (**B**) uMCP-1 [pg/mL]; (**C**) uRANTES [pg/mL]; (**D**) MCP-1/cr. [pg/mg cr.]; (**E**) RANTES/cr. [pg/mg cr.].

**Figure 2 jcm-10-00497-f002:**
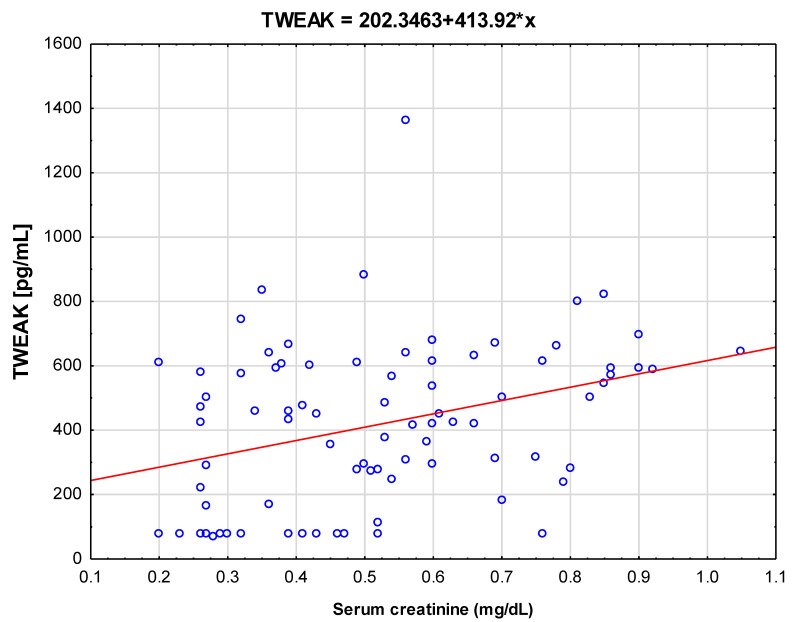
Correlations between serum tumor necrosis factor-like weak inducer of apoptosis (TWEAK) and serum creatinine levels in the study group (B).

**Figure 3 jcm-10-00497-f003:**
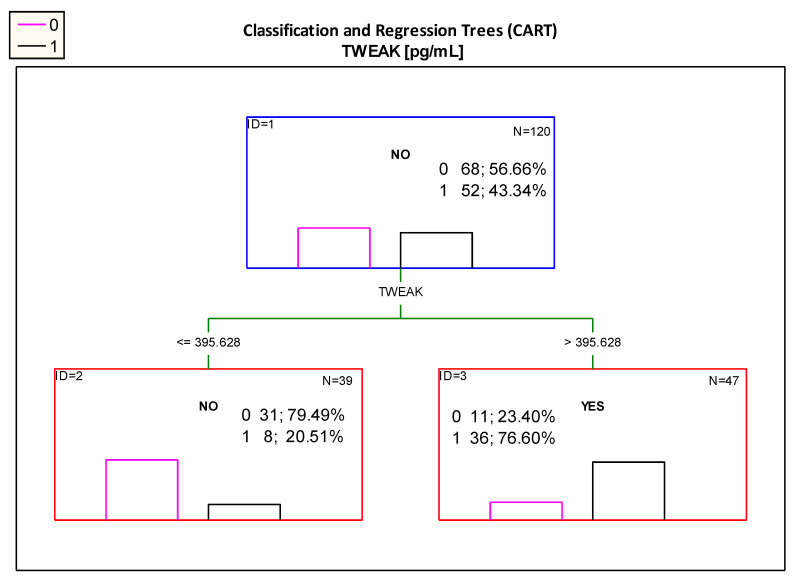
Classification tree for impaired renal function (albuminuria and/or decreased eGFR < 90 mL/min/1.73 m^2^ and/or hypertension) created by analysis taking into account the TWEAK serum concentration. Figure legend: 0—normal kidney function (pink column), 1—impaired renal function (albuminuria and/ or decreased eGFR < 90 mL/min/1.73 m^2^ and/ or hypertension) (black column).

**Figure 4 jcm-10-00497-f004:**
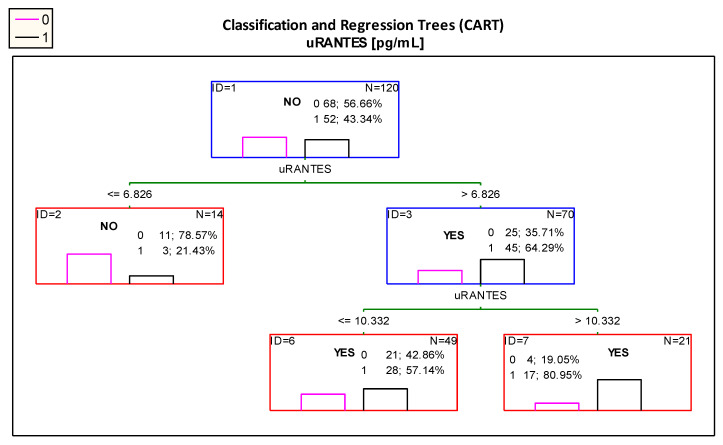
Classification tree for impaired renal function (albuminuria and/or decreased eGFR < 90 mL/min/1.73 m^2^ and/or hypertension) created by analysis taking into account the uRANTES urinary excretion. Figure legend: 0—normal kidney function (pink column), 1—impaired renal function (albuminuria and/or decreased eGFR < 90 mL/min/1.73 m^2^ and/or hypertension) (black column).

**Table 1 jcm-10-00497-t001:** Demographic, anthropometric, biochemical parameters, and estimated biomarkers in patients with solitary functioning kidney (groups A and N) and healthy peers; comparisons between estimated groups.

	**A *n* = 54** M/F: 33/21	**N *n* = 26** M/F: 15/11	**K *n* = 40** M/F: 21/19	***p*** **(A vs. N)**	***p*** **(A vs. K)**	***p*** **(N vs. K)**
	**Median** **(Q1–Q3)**					
Age (years)	9.37 (4.75–13.25)	10.16 (3.25–15.16)	6.54 (3.50–11.20)	NS	NS	NS
Body weight (kg)	33.0 (24.0–55.8)	27.0 (14.9–61.2)	24.5 (14.50–52.0)	NS	NS	NS
Height (cm)	142.0 (115.0–163.0)	145.5 (99.0–168.0)	139.0 (105.0–154.5)	NS	NS	NS
SBP (mmHg)	111.5 (102.0–120.0)	106.50 (99.00–-114.0)	100.0 (88.0–113.0)	NS	*0.022*	NS
Serum creatinine (mg/dL)	0.52 (0.39–0.63)	0.50 (0.32–0.85)	0.40 (0.30–0.60)	NS	*0.041*	NS
Serum urea (mg/dL)	27.0 (22.0–30.0)	26.0 (20.0–28.0)	29.0 (19.0–30.0)	NS	NS	NS
eGFR by Schwartz (mL/min/1.73 m^2^)	112.19 (96.81–125.37)	105.83 (89.56–128.32)	112.49 (101.12–143.92)	NS	NS	NS
Creatinine clearance (mL/min)	86.0 (62.73–115.81)	92.9 (72.26–112.0)	101.01 (119.22–152.97)	NS	NS	NS
Albuminuria (mg/day)	34.45 (8.47–92.65)	18.87 (3.26–89.21)	-	NS	-	-
UACR (mg/g cr.)	96.15 (0.0–214.66)	84.61 (0.0–-825.0)	-	NS	-	-
TWEAK (pg/mL)	486.09 (363.65–631.0)	577.18 (356.26–603.25)	164.94 (78.1–278.53)	NS	*<0.0001*	*<0.0001*
uMCP-1 (pg/mL)	50.84 (12.29–118.28)	16.61 (8.28–81.33)	12.55 (3.9–36.23)	NS	*0.01*	NS
MCP-1/cr. (pg/mg cr.)	61.14 (14.26–205.79)	61.91 (10.18–145.22)	14.52 (6.37–35.11)	NS	*0.014*	NS
uRANTES (pg/mL)	8.58 (7.23–14.07)	8.31 (7.77–9.66)	6.96 (6.42–7.50)	NS	*0.001*	*0.0004*
RANTES/cr. (pg/mg cr.)	11.90 (7.28–26.30)	15.11 (13.13–22.45)	6.49 (3.4–8.20)	NS	*0.006*	*0.004*

A—congenital unilateral renal agenesis; N—acquired SFK (after surgical removal); Q1—lower quartile; Q3—upper quartile, NS—not significant, M—males, F—females.

**Table 2 jcm-10-00497-t002:** Receiver Operating Characteristic (ROC) analyses for TWEAK, uMCP-1, MCP-1/cr., uRANTES, RANTES/cr. levels in (**A**) children with SFK among all examined children, and (**B**) in children with impaired renal function (albuminuria and/or decreased eGFR <90 mL/min/1.73 m^2^ and/or hypertension) among all SFK children.

(**A**)							
**Cut-Off Values**	**AUC**	**SE**	**−95% CI**	**+95% CI**	***p* Value**	**Sensitivity**	**Specificity**
TWEAK ≥ 288.24	0.853	0.049	0.758	0.948	<0.001	80%	91.3%
uMCP-1 ≥ 5.053	0.656	0.066	0.527	0.784	<0.05	89.2%	60.9%
MCP-1/cr. ≥ 8.907	0.670	0.062	0.549	0.791	<0.01	60%	78.3%
uRANTES ≥ 7.236	0.757	0.057	0.645	0.868	<0.001	81.5%	56.5%
RANTES/cr. ≥ 4.049	0.816	0.072	0.675	0.958	<0.001	63.8%	87.5%
(**B**)							
**Cut-Off Values**	**AUC**	**SE**	**−95% CI**	**+95% CI**	***p* Value**	**Sensitivity**	**Specificity**
TWEAK ≥ 421.934	0.790	0.050	0.691	0.889	<0.001	79.1%	76.7%
uMCP-1 ≥ 36.717	0.594	0.063	0.471	0.717	NS	57.4%	67.7%
MCP-1/cr. ≥ 48.606	0.599	0.063	0.477	0.722	NS	61.7%	67.6%
uRANTES ≥ 10.734	0.660	0.060	0.542	0.777	<0.01	34%	86.5%
RANTES/cr. ≥ 15.801	0.631	0.080	0.473	0.788	NS	40.5%	83.3%

AUC—Area Under Curve, SE—Standard Error.

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
