# Peer review of "Tumor Necrosis Factor-Like Weak Inducer of Apoptosis and Selected Cytokines—Potential Biomarkers in Children with Solitary Functioning Kidney"

_jcm, 2021, doi:10.3390/jcm10030497_

Round 1

Reviewer 1 Report

The manuscript addresses a quite interesting topic. The authors have enrolled a pretty large cohort of pediatric patients with solitary kidney ( either congenital and acquired solitary kidney) and have measured serum TWEAK and urinary cytokines in patients and controls.

I have the following suggestions to improve the strenghts of the results:

  • the study group consists of a quite etherogenous cohort of individuals: children with acquired and congenital solitary kidney, with a wide range of eGFR (from nearly 60 ml/min to over 100 ml/min) that have been compared with patients with normal renal function. I am not sure that individuals with normal renal functions represent the right control group.  Please state clearly the main aim of the manuscript :  is the identification of biomarkers of solitary kidney ( either acquired and congenital?) or of eGFR decline in general?: if the main aim is the firs one,  to improve the specificity of results a larger control group would be studied, including patients with a wider range of eGFR ( from 60 to 120 ml/min):in these two cohorts ( study cohort and controls) you should plot urinary and plasma levels of measured biomarkers with the eGFR/ albuminuria etc:  this will tell us whehter the decline of the eGFR accounts for changes in urinary and serum  measured biomarkers. If you use your cohort as a model of eGFR decline it should be specified.  
  • figure 1 shows that the behaviour of studied molecules is similar in patients with acquired and congenital disorders; figure 2 confirms this theory for TWEAK. Is the molecule accumulating due to eGFR decline? please comment. 
  • AS creatinine is known as as non-early marker of kidney dysfunction it would be of interest to address whether serum TWEAK levels increase before significant increase of plasma creatinine ( for example in  patients of our cohort with normal eGFR and albuminuria is TWEAK higher that in control individuals with comparable eGFR and non albuminuria?)
  • Table 1 pooled together data from patients with congenital and acquired solitary kidney. I would separate the two cohorts to address differences, as in figure 1 data from acquired and congenital disorders have been show as separated.
  • English editing is required ( please check also the title).

Author Response

I thank you in advance for the time and effort you expend considering my work. We have carefully reviewed the comments and have revised the manuscript accordingly. I enclose the respond in a point-by point fashion to the referees’ comments. Changes to the manuscript are shown in red.

We hope that you find our responses satisfactory and look forward to hearing from you in due course.

  • the study group consists of a quite etherogenous cohort of individuals: children with acquired and congenital solitary kidney, with a wide range of eGFR (from nearly 60 ml/min to over 100 ml/min) that have been compared with patients with normal renal function. I am not sure that individuals with normal renal functions represent the right control group.  Please state clearly the main aim of the manuscript :  is the identification of biomarkers of solitary kidney ( either acquired and congenital?) or of eGFR decline in general?: if the main aim is the firs one,  to improve the specificity of results a larger control group would be studied, including patients with a wider range of eGFR ( from 60 to 120 ml/min):in these two cohorts ( study cohort and controls) you should plot urinary and plasma levels of measured biomarkers with the eGFR/ albuminuria etc:  this will tell us whehter the decline of the eGFR accounts for changes in urinary and serum  measured biomarkers. If you use your cohort as a model of eGFR decline it should be specified.  

I am not sure if I understood Reviewer’s comments correctly, but we tried to rearrange studied groups and we divided patients depending on eGFR ≤ 60ml/min and >60ml/min into study and control cohorts:

A- Congenital disorder with eGFR ≤60mL/min

AK-  Congenital disorder with eGFR: 60mL/min -120mL/min

N- acquired SFK with eGFR ≤60mL/min

NK- acquired SFK with eGFR: 60mL/min -120mL/min

K – healthy peers

However performed analysis have very low power due to the small group of children with eGFR ≤60mL/min: A - 7 children, N group - 4 participants.

Nonetheless, analysis that we performed did not revealed significant corellations in estimated markers concentrations and did not show its dependece with eGFR or albuminuria/ UACR.

Because of the small samples we were unable to draw the specific, intended conclusions. That is why we did not include it into revised paper.

  • figure 1 shows that the behaviour of studied molecules is similar in patients with acquired and congenital disorders; figure 2 confirms this theory for TWEAK. Is the molecule accumulating due to eGFR decline? please comment. 

Due to suggestions we have added comment in Discussion section regarding possible increased concentration of TWEAK due to its accumulation:

'it raised the question if the molecule is accumulating due to eGFR decline?

All of evidence suggests that TWEAK is involved in the pathological processes that occur locally in the kidneys. Indeed, it is unknown whether a relevant increase in the production of TWEAK or a severe reduction in renal filtration of TWEAK may increase their serum concentrations. Currently it is thought that TWEAK levels increase in patients with impaired renal function. However, up to now, very few data are available on the precise relationship between TWEAK and the level of GFR. In fact, in patients with severe decline in eGFR serum concentrations of different proteins are definitely increased, and as a consequence, the filtered load of these proteins to the residual nephrons may become higher than single-nephron maximal tubular reabsorptive capacity, and its urinary excretion should increase. Interestingly, the urinary excretion of molecules begins from different threshold values of GFR and at different levels of serum concentrations, indicating that proximal tubular cells probably have a different reabsorptive capacity for the different proteins. 

Further studies comparing serum TWEAK concentration with its urinary excretion would be helpful to answer this question in detail.'

  • AS creatinine is known as as non-early marker of kidney dysfunction it would be of interest to address whether serum TWEAK levels increase before significant increase of plasma creatinine ( for example in  patients of our cohort with normal eGFR and albuminuria is TWEAK higher that in control individuals with comparable eGFR and non albuminuria?)

We have checked and addressed obtained results in Results section, that TWEAK serum levels in studied children with albuminuria and normal eGFR presented significantly higher concentrations than controls (median 512.23, Q1-Q3: 389.09-731.51pg/mL, p= 0.00006).

  • Table 1 pooled together data from patients with congenital and acquired solitary kidney. I would separate the two cohorts to address differences, as in figure 1 data from acquired and congenital disorders have been show as separated.

We have seperated data regarding groups: A, N and K in Table 1.

  • English editing is required ( please check also the title).

We have tried to improve English.

Reviewer 2 Report

Hanna Nosek et al. wrote an article titled: Tumor necrosis factor-like weak inducer of apoptosis and selected cytokines as a potential biomarkers in children with solitary functioning kidney. The study is very well presented and written. However, as expressly said from the authors, this study has some limitations such as being small, single center and cross-sectional. As for my self I believe that the study present a table with the characteristics of the participants that needs to be improved showing also information about age gender weight to better define the  analysed samples. I truly think this table needs to be implemented considering the difficulties related with the analyses of GFR.

Author Response

We are grateful for your time, consideration, and precise revision. We have carefully reviewed the comments and have revised the manuscript accordingly. Changes to the manuscript are shown in red.

We hope that you find our responses satisfactory and look forward to hearing from you in due course.

As for my self I believe that the study present a table with the characteristics of the participants that needs to be improved showing also information about age gender weight to better define the  analysed samples.

We have added suggested table to include more specific data regarding studied participants.

Reviewer 3 Report

The study investigated various urinary proteins as a potential marker for early renal impairment. The authors were able to detect two markers as possible urinary biomarkers in children with solitary functioning kidney. Furthermore, with a comparison and regression analysis they were able to detect specific cut-off values, indicating the kidney impairment.

The information written in the introduction and material+ method section is clear and easy to understand. The result section should provide the reader with more references to the figures and some additional points mentioned below. Furthermore, it might be helpful for the readers if the figure legends would describe in more detail what the figure/ table represents.

  • line 93: no point 3 was mentioned
  • line 97: study included 120 children (this was then split into study group B and control group K)
  • line 106: female/male ratio was provided for the first subgroup for study group B. Therefore, provide these as well for the second subgroup (normal/ impaired kidney function).
  • line 181: values for measured systolic blood pressure as illustration could be useful for the readers.
  • line 188: this paragraph could be combined with the one from line 181 on. Also, a graphical illustration/ table might be helpful.
  • Line 198-207 and Table 1:
    • data for subgroups A and N could be helpful for the readers
    • p value for serum creatinine is for group A vs. K, whereas p value for TWEAK is for group B vs. K, and some p values for (urea, uric acid, eGFR) is for A vs.N. make clear to which (sub)groups the p value contributes.
  • Figure 1: it should be clear to which two groups the p value contributes to. Furthermore, a legend for the stars indicating p-value should be provided.
  • line 269: graphical illustration for the CART of serum TWEAK would be interesting.
  • Figure 3: first separation: N=14 and N=70. But one started with N=120? Why was the rest excluded? The values 0/1 and the colors (pink and black), respectively, should be explained in a legend in the figure.

Overall, the authors research an important and interesting topic, which could in future be very helpful for early detection of kidney impairment.

Author Response

We are grateful for your time, consideration, and precise revision. We have carefully reviewed the comments and have revised the manuscript accordingly. Changes to the manuscript are shown in red.

We hope that you find our responses satisfactory and look forward to hearing from you in due course.

  • line 93: no point 3 was mentioned

Unwilling error, now numbered correctly

  • line 97: study included 120 children (this was then split into study group B and control group K)

We have changed description according suggestion

  • line 106: female/male ratio was provided for the first subgroup for study group B. Therefore, provide these as well for the second subgroup (normal/ impaired kidney function).

We have added suggested information

  • line 181: values for measured systolic blood pressure as illustration could be useful for the readers.

We have added those data into changed Table 1.

  • line 188: this paragraph could be combined with the one from line 181 on. Also, a graphical illustration/ table might be helpful.

We combined mentioned paragraphs, and have also added those data into changed Table 1.

  • Line 198-207 and Table 1:
    • data for subgroups A and N could be helpful for the readers
    • p value for serum creatinine is for group A vs. K, whereas p value for TWEAK is for group B vs. K, and some p values for (urea, uric acid, eGFR) is for A vs.N. make clear to which (sub)groups the p value contributes.

We tried to make presented data more clear and studied group are shown in Table 1 seperated in subgroups.

  • Figure 1: it should be clear to which two groups the p value contributes to. Furthermore, a legend for the stars indicating p-value should be provided.

We tried to improve and clarify data presented in Figure 1, and we highlighted legend regarding p - value

  • line 269: graphical illustration for the CART of serum TWEAK would be interesting.

We added suggested Figure as Figure 3.

  • Figure 3: first separation: N=14 and N=70. But one started with N=120? Why was the rest excluded? The values 0/1 and the colors (pink and black), respectively, should be explained in a legend in the figure.

We have changed numeration of this Figure into Figure 4.

Firstly CART analysis was performed in all studied participants distinguishing them into those with impaired renal function (1) and without (0), first separation excluded most of controls due to lack of some parameters (i.e. albuminuria, ABPM measurements, only single control individuals presented with this data) necessary for further assessment of kidney function, that is why computer program excluded those children from further analysis.

Explanation for 0 and 1 were included into Figure title, but we try to make it more clear so we put it into Figure legend. Explanation of the colors used has been also described in Figure legend:

0 – normal kidney function (pink column)

1 – impaired renal function (albuminuria and/ or decreased eGFR <90 mL/min/1.73m2 and/ or hypertension) (black column)

Round 2

Reviewer 1 Report

The manuscript is improved. Limitations have been addressed in the discussion. 

No additional requirements.